# Oxidative Stress-Mediated DNA Damage Induced by Ionizing Radiation in Modern Computed Tomography: Evidence for Antioxidant-Based Radioprotective Strategies

**DOI:** 10.3390/antiox14091085

**Published:** 2025-09-04

**Authors:** Baltasar Ramos, Jorge Gómez-Cayupán, Isidora Aranis, Esperanza García Tapia, Constanza Coghlan, María-José Ulloa, Santiago Gelerstein Claro, Katherine Urbina, Gonzalo Espinoza, José De Grazia, Jorge Díaz, Prisco Piscitelli, Daniel Castro, Marcia Manterola, Ramón Rodrigo, Camilo G. Sotomayor

**Affiliations:** 1Radiology Department, University of Chile Clinical Hospital, Carlos Lorca 999, Independencia, Santiago 8380453, Chile; baltasarramos@ug.uchile.cl (B.R.); mulloaj@udd.cl (M.-J.U.); gespinozao@hcuch.cl (G.E.); jdegrazia@hcuch.cl (J.D.G.); jorgediaz@uchile.cl (J.D.); dcastro@hcuch.cl (D.C.); 2Human Genetics Program, Institute of Biomedical Sciences, Faculty of Medicine, Universidad de Chile, Independencia, Santiago 8380453, Chile; jorge.gomez.c@uchile.cl (J.G.-C.); katherine.urbina@ug.uchile.cl (K.U.); mmanterola@uchile.cl (M.M.); 3School of Medicine, Faculty of Medicine, University of Chile, Independencia, Santiago 8380453, Chile; isidora.aranis@ug.uchile.cl (I.A.); esperanza.garcia@ug.uchile.cl (E.G.T.); constanza.coghlan@ug.uchile.cl (C.C.); s.gelerstein2002@gmail.com (S.G.C.); 4Department of Psychology and Health Sciences, Pegaso University, Piazza Trieste e Trento 48, 80132 Naples, Italy; prisco.piscitelli@unipegaso.it; 5Laboratory of Oxidative Stress and Nephrotoxicity, Molecular and Clinical Pharmacology Program, Institute of Biomedical Sciences, Faculty of Medicine, University of Chile, Independencia, Santiago 8380453, Chile; rrodrigo@uchile.cl; 6Faculty of Medicine, San Sebastián University, Campus Los Leones, Lota 2465, Providencia, Santiago 7510157, Chile

**Keywords:** computed tomography, ionizing radiation, oxidative stress, DNA damage, antioxidants

## Abstract

Computed tomography (CT) is fundamental to modern medicine, yet ionizing radiation (IR) exposure causes DNA damage. Although often underestimated, at current doses, CT may account for ~5% of new cancer diagnoses. Complementary radioprotective approaches beyond dose reduction are needed. We conducted a prospective observational study to characterize IR-induced oxidative stress (OS)-mediated DNA damage in modern CT to explore potential antioxidant-based radioprotective strategies. In volunteers not exposed to IR (A_NONE_) and in patients with two-phase abdominal–pelvis CT (B_EXPOSURE_), blood samples were collected at T_BASE_-min 0 and T_POST_-min 60 to measure biomarkers of OS (oxidative damage and antioxidant capacity) and DNA damage. Thirty-five subjects (*n* = 17 A_NONE_/18 B_EXPOSURE_) were studied. Body mass index and DNA damage in T_BASE_ were comparable between groups. In A_NONE_, biomarkers of OS and DNA damage did not change between T_BASE_ and T_POST_ (*p* > 0.05 for all). In B_EXPOSURE_, DNA damage was significantly increased [15% (−15–60); *p* < 0.001], which was associated with consistent increased antioxidant enzyme activity [*p* < 0.05 for all antioxidant enzymes]. In modern CT with relatively low effective dose (ED) levels, a significant increase in DNA damage was observed along with increased antioxidant enzyme activity as defensive response and marker of OS-mediated damage-mediating mechanisms. These findings warrant interventional studies to evaluate antioxidant-based radioprotective strategies.

## 1. Introduction

Computed tomography (CT) is fundamental to modern medicine and is among the most widely used imaging examinations. However, total human radiation exposure has increased significantly. While natural background radiation exposure is approximately 3 mSv per year, a whole-body CT scan provides approximately 10 mSv [1,2]. In particular, two-phase contrast-enhanced abdominal–pelvis protocol is the CT examination that contributes the most to the collective medical dose worldwide [3,4]. Despite the steady increase in CT use, the potential biological damage is often underestimated in current clinical practice, probably due to achievements in dose reduction with modern CT equipment and protocols. However, it has recently been proposed that, if current radiation dosing practices persist, CT could represent up to a 5% cancer burden, comparable to other major risk factors such as alcohol consumption and overweight [5].

For the association between human radiation exposure and DNA damage and, ultimately, risk of cancer, a direct route involving particle or photon–DNA collisions was first proposed as a damage pathway [6]. Later, an indirect mechanism of IR-induced DNA damage mediated by oxidative stress (OS) was described, which accounts for the majority, on the order of 60–70%, of the DNA damage induced by ionizing radiation overall in low linear energy transfer radiation such as CT scans [7,8,9].

Oxidative stress arises from an asymmetry in which the production of oxidative species outpaces the antioxidant enzyme defense system activity, favoring the former. The indirect mechanism mediated by OS was described as the formation of reactive oxygen species (ROS) resulting from the radiolysis of water triggered by IR. If these species are not efficiently neutralized, they induce structural chromosomal aberrations, potentially initiating mutations and thereby contributing to cancer development [10,11,12].

Understanding these mechanisms, particularly the indirect mechanism mediated by the OS of IR-induced DNA in modern CT, is clinically relevant, as it could open an underexplored avenue toward novel antioxidant-based preventive strategies to mitigate CT-related DNA damage. Currently, clinical studies evaluating the role of OS in the association between radiation exposure and DNA damage are lacking. To subsequently propose an antioxidant-based radioprotective strategy, we aimed to characterize OS and IR-induced DNA damage in modern CT.

## 2. Materials and Methods

A prospective observational study was performed in the Radiology Department of the University of Chile Clinical Hospital between April and August 2024. This study was approved by the Medical Ethics Committee (AA 84/23 on 10 January 2024) and the hospital administration, and it received certification from the Clinical Research Office (AC 1399/24 on 25 January 2024). The full study protocol has been previously published [13].

### 2.1. Participants

Volunteers from healthcare personnel not exposed to IR (and their relatives) with no indication for IR-associated testing (A_NONE_ group) were recruited through direct invitation from the researchers. Consecutive patients with abdominal or pelvic organ disease who had a clinical indication for two-phase contrast enhanced (CE) abdominal–pelvis CT were also recruited through the Outpatient CT Schedule (B_EXPOSURE_ group). All study subjects (men and women) were between 18 and 75-year-old and able to provide a written informed consent. Women who were pregnant and patients with advanced chronic kidney disease defined as glomerular filtration rate < 30 mL/min/1.73 m^2^, or patients with known contraindications to antioxidant supplements or to the use of iodinated contrast agents were excluded. Subjects with genetic syndromes, onco-hematologic diseases or a history of peptic ulcers or urinary stones, occupational exposure to other IR-related, radiation therapy or chemotherapy within the previous 6 months, or exposure to other IR-related studies within the 72 h prior to or immediately after abdominal–pelvic CT were also excluded. Subjects who used premedication for CE CT, regular use of antioxidant supplements or intake on the day of the CT, iron supplements, or who followed a low-iron diet were also excluded.

In both groups, blood sample #1 (T_BASE_) was collected at minute 0, and blood sample #2 (T_POST_) approximately at minute 60 (±10). A clinical interview (T_CI_) was conducted between the two groups. In B_EXPOSURE_, the CT scan was scheduled approximately 5 min before blood sample #2 (Figure 1).

### 2.2. CT Study Protocol

All patients were scanned on first-generation dual-energy CT systems (SOMATOM Definition Edge, Siemens Healthineers, Erlangen, Germany). Scan range extended from the diaphragm to the pubic symphysis. CT parameters were: tube voltage, 100 kVp (standard protocol); effective tube current, 180 mAs (standard protocol); rotation time, 0.5 s. Following the non-contrast acquisition, a non-ionic contrast agent was administered intravenously (Appendix A).

### 2.3. Ionizing Radiation Dose Estimation

Ionizing radiation dose was estimated based on the patient-specific effective dose (ED), in accordance with ICRP Publication 103 (International Commission on Radiological Protection) [14] and the American Association of Physicists in Medicine (AAPM) Reports No. 204 and No. 220 [15] based on the calculated Dose-Length Product (DLP), the corresponding body segment conversion factor (0.014 for abdomen and pelvis), and the effective diameter. The effective diameter (anteroposterior + lateral) was measured from a transverse CT image at the level of the L3 spine.

### 2.4. DNA Damage Determination

DNA damage was determined by γ-H2AX foci quantification in peripheral lymphocytes through immunofluorescence. Isolated peripheral lymphocytes incubated with mouse anti-γ-H2AX antibody diluted 1:100 in PBS, anti-mouse Cy2 secondary antibody diluted 1:300. and counterstained with 4’-6-diamidino-2-phenylindol (DAPI).

### 2.5. Determinations of Oxidative Stress-Related Parameters

Oxidative stress was assessed by measurements of plasma F2-isoprostane levels, as these biomarkers are products of nonenzymatic peroxidation of arachidonic acid. This determination is based on a competitive enzyme-immunoassay that quantifies 8-iso-prostaglandin F_2_α by incubating plasma samples with a specific antibody and an acetylcholinesterase-conjugated tracer in antibody-coated wells at 4 °C. Ellman’s color reagent and a small tracer boost were added, and absorbance at 405–420 nm. Values were expressed as pg mL^−1^. Total plasma antioxidant capacity was assessed spectrophotometrically by measuring the ability to reduce Fe^3+^ to Fe^2+^ with an acidic ferric 2,4,6-tripyridyl-s-triazine (TPTZ), forming a blue Fe^2+^-TPTZ complex that absorbs at 593 nm. Erythrocyte catalase activity was assayed from the kinetic of breakdown of hydrogen peroxide at 240 nm by the erythrocyte supernatant of 2400 g; it was expressed on the basis of the rate constant of the first order reaction (k)/mg protein. Superoxide dismutase (SOD) assay is based on the inhibition of the auto-oxidation of epinephrine to adrenochrome at pH 10.2, monitored as a reduced rate of absorbance increase at 480 nm; one SOD unit is defined as the activity that reduces to the half the autooxidation background; SOD activity is expressed as units/mg protein. Soluble glutathione peroxidase (GSH-Px) activity was measured in the cytosolic fraction (100,000 g supernatant) by a spectrophotometric method based on the reduction of glutathione disulfide coupled to the NADPH oxidation by glutathione reductase. One GSH-Px unit is defined as the activity that oxidizes 1 µmol NADPH per minute. The activity of GSH-Px was expressed as U/mg protein. 

### 2.6. Data Storage

The data was anonymized and stored on SASIBA, a cloud-based data platform with access restricted to approved researchers. SASIBA is operated by the Data Unit of the Center for Medical Informatics and Telemedicine at the Faculty of Medicine and is available to the local scientific community. The platform is hosted within the University of Chile’s central data center and is protected by the same safeguards applied to the University’s confidential data, including physical security policies aligned with Tier II/III standards. 

### 2.7. Statistical Analyses

Continuous variables were summarized as mean ± SD when normally distributed and as median (interquartile range) otherwise. Categorical variables were summarized as percentages. Cross-sectional differences between groups at each time point (T_BASE_ and T_POST_) were assessed with Student’s *t* test or the Mann–Whitney U test for normally and non-normally distributed variables, respectively. Within-group longitudinal changes were evaluated with the paired Student’s *t* test or the Wilcoxon signed-rank test. Box and whisker plots were generated to depict distributional differences in overall survival (OS) and DNA-damage biomarkers among exposed participants. To quantify associations multivariable linear-regression analyses were performed. Across all analyses, *p* values < 0.05 were deemed statistically significant. Data were analyzed with IBM SPSS version 29 (SPSS Inc., Chicago, IL, USA), STATA 18 (STATA Corp., College Station, TX, USA), and R version 4.3.1 (R Foundation for Statistical Computing, Vienna, Austria).

## 3. Results

Thirty-five subjects were included, *n* = 17 and 18 for the A_NONE_ and B_EXPOSURE_, respectively. Age and sex were different between groups. Body mass index (BMI) and T_BASE_ DNA damage were not significantly different between groups. Table 1 presents detailed summary of baseline characteristics and study-group comparisons.

A baseline comparison analyses (T_BASE_) between groups regarding OS biomarkers (Table 2) showed higher levels of the antioxidant enzymes glutathione peroxidase (*p* = 0.03), superoxide dismutase (*p* < 0.001) and catalase activity (*p* < 0.001) in healthy volunteers (A_NONE_), with no significant differences in lipid peroxidation and antioxidant capacity status by F2-isoprostanes (*p* = 0.72) and FRAP (*p* = 0.38), respectively.

During T_EXP_ (B_EXPOSURE_ group only), the non-CE and CE phases of the CT examination effectively occurred 11 (3, 16) and 8 (1, 9) minutes before T_POST_. DLP and patient-specific ED was 381 (347, 571) mGy*cm and 6.7 (6.2, 9.0) mSv, respectively (Appendix A).

A relative increase in the activities of antioxidant enzymes after exposure (in the B_EXPOSURE_ group) was observed (Table 3). At a difference with T_BASE_, glutathione peroxidase (*p* < 0.15) and superoxide dismutase (*p* < 0.28) levels in the patient group did not differ from those in the healthy volunteer group, only the remaining catalase activity was significantly lower [1241 (976, 1541) and 1398 (1188, 1821), respectively for the B_EXPOSURE_ and A_NONE_ groups; *p* = 0.02]. Similar to the comparative T_BASE_ analyses, no significant differences in lipid peroxidation were observed between groups at this time point (*p* = 0.29). The trend towards increased FRAP in the B_EXPOSURE_ group did not result in significant differences between groups (*p* = 0.19). Increased DNA damage in the patient group resulted in a significant between-group difference [1.8 ± 0.8 and 1.3 ± 0.2 respectively, for the B_EXPOSURE_ and A_NONE_ groups; *p* = 0.02], in contrast to the comparative T_BASE_ results [1.3 ± 0.2 and 1.4 ± 0.5, respectively, for the B_EXPOSURE_ and A_NONE_ groups; *p* = 0.42]. To compare longitudinal changes between T_BASE_ and T_POST_ (Table 4), absolute and relative changes in biomarkers of OS and DNA damage are presented. Results from within-group longitudinal change tests are also presented. Analyses show that between T_BASE_ and T_POST_, biomarkers of OS and DNA damage did not change over time in the A_NONE_ control group of healthy volunteers (*p* > 0.05 for all biomarkers). On the other hand, in the B_EXPOSURE_ group, DNA damage significantly increased [15% (−15, 60); 40% ± 72; *p* < 0.001]. The same pattern was observed in the comparative analyses of change between groups, where significant differences were found in both absolute and relative change compared to the A_NONE_ group (*p* = 0.02 and *p* = 0.02, respectively). Similarly, a consistent increase in the activities of antioxidant enzymes was found [10% (0, 15), 15% (−2, 38) and 26% (5, 80) for glutathione peroxidase, superoxide dismutase and catalase activity, respectively; *p* < 0.05 for all] (Figure 2). This was also supported by the results of the between-group change comparison analyses, in which significant differences were found in absolute and relative changes compared with group A_NONE_ on the superoxide dismutase (*p* < 0.001 and *p* < 0.001, respectively) and catalase (*p* < 0.001 and *p* < 0.001, respectively) activities.

Further analyses were performed to explore whether baseline variables were associated with the change in DNA damage. Multiple linear-regression analyses showed that there is no significant association of age and sex with γ-H2AX change (T_POST_ − T_BASE_) in the overall population, neither in the A_NONE_ nor B_EXPOSURE_ groups (Table 5).

## 4. Discussion

Overall, the B_EXPOSURE_ group showed a significant, absolute and relative increase in DNA damage, accompanied by higher antioxidant-enzyme activities, whereas these parameters did not show any significant changes in the A_NONE_ group. This observational study focuses on DNA damage produced by the single diagnostic CT that delivers the highest cumulative global radiation dose: contrast-enhanced abdominal–pelvis CT [16]. Apart from previous in vitro or mixed designs [17], this is, to our knowledge, the first outpatient study explicitly designed to test the hypothesis that ROS substantially mediate CT-related, ionizing-radiation (IR)–induced DNA damage.

Building on Tao et al. [18], we incorporated patient-specific dose estimates, allowing comparison with future work in other IR-based imaging modalities according to international risk-estimation guidelines and a further strength of this study. An additional strength of this study is the inclusion of an exposed cohort and an unexposed control group, each adequately powered for statistical analysis.

At our center, biphasic CE abdominal–pelvis CT delivers lower effective doses (EDs) than most published series. For example, Tao et al. [18] reported a mean DLP of 1358.7 mGy·cm, three to four times higher than that of our exposed cohort. This is consistent with the downward trend in CT dose estimates (current mean ED ≈ 7.7 mSv) described by Mettler et al. [16], suggesting that our findings reflect contemporary equipment and protocols. Despite these lower doses, we observed a significant increased DNA damage both longitudinally (pre- and post-scan) and cross-sectionally (exposed vs. controls). The absolute and relative increases are consistent with those of Tao et al. and fit the low- and high-dose data of Stehli et al. [19] Similarly, Rothkamm et al. [20] linked ~0.24 γ-H2AX foci cell^−1^ to ~6.3 mGy, which is consistent with our dose and damage estimates underscoring the need for additional protective measures even at current CT dose levels. The simultaneous upregulation of enzymatic antioxidants favors an IR-induced ROS mechanism. Considering that nuclear factor erythroid 2–related factor 2 (Nrf2) regulates a broad antioxidant response [21,22], our data suggest that supplementation with exogenous antioxidants could help mitigate CT-related DNA damage, similar to the cardioprotective benefit demonstrated in cardiovascular disease [23].

From a methodological standpoint, we scheduled the T_POST_ blood draw 60 min after T_BASE_ to align with our interventional study; this timing mirrors our ongoing randomized, double-blind, placebo-controlled trial, in which participants receive oral vitamin C 60 min before abdominopelvic CT [13]. Using the same window improves comparability of effect sizes and reduces timing-related confounding between the observational and interventional cohorts.

We also considered potential confounding factors related to participant demographic characteristics. The B_EXPOSURE_ participants were older and predominantly female in comparison to A_NONE_. Because baseline differences in age and sex between the A_NONE_ and B_EXPOSURE_ groups may, in theory, influence radiosensitivity and oxidative stress responses [24,25,26] we analyzed whether these demographic variables were associated with the variation in γ-H2AX between T_BASE_ and T_POST_. These analyses showed that there is not significant association of age and sex with γ-H2AX change (T_POST_ − T_BASE_) in the overall population, nor in analyses by A_NONE_ and B_EXPOSURE_ groups. In line with current standards, ICRP 103 [14] adjusts the ED only for the pediatric versus adult population, not for more specific age or sex strata in adults, limiting the impact of these differences.

Limitations include reliance on a single CT protocol and only γ-H2AX because it was the only technically and financially feasible assay in our laboratory. We did not measure upstream ATM/ATR activation, p53 phosphorylation, or caspase activity which rise in parallel with γ-H2AX after IR [27,28,29,30]. As a result, we cannot delineate the full cascade from lesion sensing to apoptosis; future studies should incorporate these markers. Another important limitation of this stage is the limited statistical power, which constrained our ability to fit more complex multivariable/multivariate models and to characterize the full complexity of the process. The modest sample size increases the risks of type II error and overfitting; accordingly, the absence of clear evidence of an effect warrants cautious interpretation. Although this study was aimed at analyzing the immediate mechanisms of OS, the ultimate goal is to clarify whether mitigating these early effects translates into a reduction in long-term consequences, such as cancer. Related to this issue, there is ongoing concern about CT, stemming from reports of cancer risk and its overuse [5,31,32]. Even in institutions with decision-support systems and doses lower than the national reference doses, large-scale data rekindles these concerns. Therefore, it is essential to focus on improving patients’ intrinsic defenses against IR, particularly their antioxidant capacity.

Vitamin C emerges as a promising radioprotective strategy against radiation-induced oxidative stress. It is among the most potent antioxidants: it directly scavenges ROS by donating electrons and oxidizing to dehydroascorbate; it also indirectly limits ROS generation by inhibiting pro-oxidant enzymes and the NF-κB pathway, stabilizing tetrahydrobiopterin, and regenerating α-tocopherol [33,34,35,36,37,38,39,40,41]. Evidence from in vitro, in vivo and mixed studies supports its use to lessen CT-induced DNA damage [17,18,42,43,44,45]. Across tested agents, vitamin C and N-acetylcysteine (NAC) achieve the largest reductions, with vitamin C showing roughly twice the protective effect of NAC in clinical settings [19]; however, combinations of antioxidants have not surpassed vitamin C as a single agent [43]. An intermediate oral dose of vitamin C (~1 g) given about 60 min before RI exposure has been linked to approximately 61% lower DNA damage [18]. Encouraged by these findings, we are conducting a prospective interventional randomized placebo-controlled study under blind conditions at the University of Chile Clinical Hospital in order to assess the benefits of vitamin C administered prior to IR exposure from abdominal and pelvic CT.

## 5. Conclusions

In modern CT, at relatively low levels of ED, we observed a significant increase in DNA damage, along with increased antioxidant enzyme activity as a defensive response and a marker of OS in damage-mediating mechanisms. These findings justify interventional studies to evaluate a potential antioxidant-based radioprotective strategy.

## Figures and Tables

**Figure 1 antioxidants-14-01085-f001:**
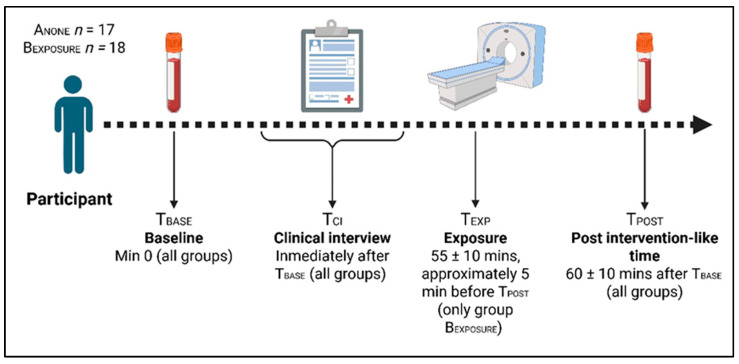
Study protocol timeline. A_NONE_ (*n* = 17), B_EXPOSURE_ (*n* = 18). T_BASE_, baseline, informed consent signing and collection of blood sample #1 at min 0 (all groups). T_CI_, clinical interview, immediately after T_BASE_ (all groups). T_EXP_, exposure, at min 55 ± 10 (only group B_EXPOSURE_), approximately 5 min before T_POST_. T_POST_, post-exposure, collection of blood sample #2, at min ~60 ± 10 min after T_BASE_ (all groups).

**Figure 2 antioxidants-14-01085-f002:**
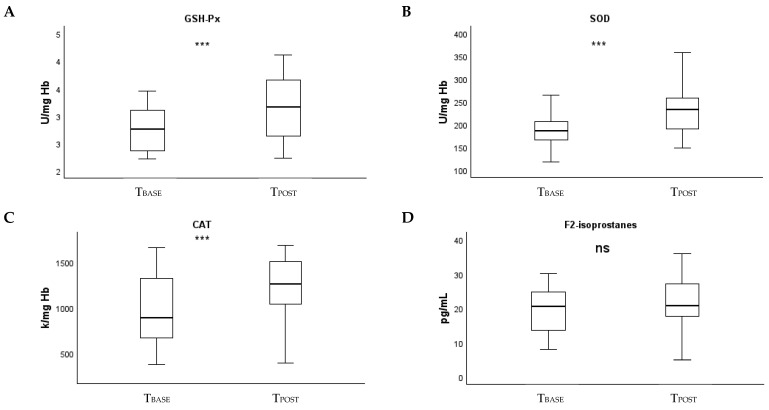
Absolute biomarkers changes (T_POST_ − T_BASE_) in B_EXPOSURE_ group. A *p* value < 0.05 was considered statistically significant. ns, non-significant; ***, significant. (**A**), GSH-Px; (**B**), SOD; (**C**), CAT; (**D**), F2-isoprostanes; (**E**), FRAP; (**F**) γ-H2AX (* and ° represent outlier instances). CAT, catalase activity; FRAP, ferric reducing ability of plasma; GSH-Px, glutathione peroxidase; Hb, hemoglobin; SOD, superoxide dismutase.

**Table 1 antioxidants-14-01085-t001:** Demographic characteristics of groups A_NONE_ and B_EXPOSURE_ at baseline (T_BASE_).

Characteristics	A_NONE_	B_EXPOSURE_	*p*
	*n* = 17	*n* = 18	
Age, years	30 ± 12	55 ± 13	<0.001
Gender, female	5 (30)	15 (83)	0.001
BMI, kg/m^2^	24.0 (22.5, 28.2)	26.5 (23.0, 31.5)	0.30
Comorbidity			-
Benign abdominal lesions	-	7 (39)	
Abdominal infection/inflammation	-	4 (22)	
Abdominal wall hernia	-	2 (11)	
Other	-	5 (28)	
γ-H2AX, foci/cell	1.4 ± 0.5	1.3 ± 0.2	0.42

Values are presented as mean ± SD, median (interquartile range), or *n* (%). Cross-sectional differences between groups were assessed using Student’s *t* test or the Mann–Whitney U test for normally and non-normally distributed continuous variables, respectively, and the chi-square test for categorical variables. A *p* value < 0.05 was considered statistically significant. BMI, body mass index.

**Table 2 antioxidants-14-01085-t002:** Oxidative stress biomarkers at T_BASE_ (baseline).

Biomarker	A_NONE_	B_EXPOSURE_	*p*
	*n* = 17	*n* = 18	
GSH-Px, U/mg Hb	3.1 ± 0.5	2.8 ± 0.4	0.03
SOD, U/mg Hb	272 ± 56	197 ± 39	<0.001
CAT, k/mg Hb	1565 ± 604	905 ± 354	<0.001
F2-isoprostanes, pg/mL	20 ± 2	20 ± 7	0.72
FRAP, µmol/L	356 (277, 469)	305 (265, 408)	0.38

Values are presented as mean ± SD or median (interquartile range). Cross-sectional differences between groups were assessed using Student’s *t* test or the Mann–Whitney U test for normally and non-normally distributed continuous variables, respectively. A *p* value < 0.05 was considered statistically significant. CAT, catalase activity; FRAP, ferric reducing ability of plasma; GSH-Px, glutathione peroxidase; Hb, hemoglobin; SOD, superoxide dismutase.

**Table 3 antioxidants-14-01085-t003:** Oxidative stress and DNA damage biomarkers at T_POST_ (~60 min after T_BASE_).

Biomarker	A_NONE_		B_EXPOSURE_		*p*
	*n* = 17		*n* = 18		
GSH-Px, U/mg Hb	3.2	(2.9, 3.6)	3.0	(2.6, 3.5)	0.15
SOD, U/mg Hb	242	(232, 268)	232	(195, 262)	0.28
CAT, k/mg Hb	1398	(1188, 1821)	1241	(976, 1541)	0.02
F2-isoprostanes, pg/mL	22.9	(20.4, 27.8)	20.9	(17.6, 26.1)	0.29
FRAP, µmol/L	405	(267, 472)	297	(272, 422)	0.19
γ-H2AX, foci/cell	1.3	± 0.2	1.8	± 0.8	0.03

Values are presented as mean ± SD or median (interquartile range). Cross-sectional differences between groups were assessed using Student’s *t* test or the Mann–Whitney U test for normally and non-normally distributed continuous variables, respectively. A *p* value < 0.05 was considered statistically significant. CAT, catalase activity; FRAP, ferric reducing ability of plasma; GSH-Px, glutathione peroxidase; Hb, hemoglobin; SOD, superoxide dismutase.

**Table 4 antioxidants-14-01085-t004:** Longitudinal change in oxidative stress and DNA damage biomarkers between T_POST_ and T_BASE_.

Biomarker	A_NONE_			B_EXPOSURE_			*p*
	*p* ^¥^	*n* = 17		*p* ^¥^	*n* = 18		
GSH-Px, change	ns			***			
Absolute, U/mg Hb		0.2	(−0.3, 0.3)		0.2	(0.0, 0.4)	0.17
Relative, %		5	(−9, 12)		10	(0, 15)	0.21
SOD, change	ns			***			
Absolute, U/mg Hb		−22	(−42, 7)		30	(−4, 71)	<0.001
Relative, %		−8	(−12, 3)		15	(−2, 38)	<0.001
CAT, change	ns			***			
Absolute, k/mg Hb		40	(238, 207)		262	(37, 652)	0.01
Relative, %		3	(−12, 15)		26	(5, 80)	0.01
F2-isoprostanes, change	ns			ns			
Absolute, pg/mL		2.4	(−0.9, 7.3)		2.1	(−1.9, 5.7)	0.40
Relative, %		15	(−5, 30)		11	(−11, 33)	0.35
FRAP, change	ns			ns			
Absolute, µmol/L		8	(−19, 27)		14	(−30, 61)	0.42
Relative, %		2	(−4, 7)		5	(−8, 17)	0.29
γ-H2AX, change	ns			***			
Absolute, foci/cell		0.0	(−0.4, 0.3)		0.2	(−0.2, 0.7)	0.02
Relative, %		−2	(−25, 26)		15	(−15, 60)	0.02

For simplicity purposes only, in this table, independent of distribution, values are presented as median (interquartile range). For each biomarker, the absolute change was calculated as the subtraction of levels at T_POST_ minus levels at T_BASE_. For each biomarker, the relative change was calculated as the division of the absolute change by T_BASE_ levels, multiplied by 100. Cross-sectional differences between groups were assessed using Student’s *t* test or the Mann–Whitney U test for normally and non-normally distributed continuous variables, respectively. A *p* value < 0.05 was considered statistically significant. Also, paired Student’s *t* test and paired Mann–Whitney U test—were performed to investigate significant individual changes within each group. The results of the latter analyses are reported under the columns *p* ¥ (ns, non-significant; ***, significant). CAT, catalase activity; FRAP, ferric reducing ability of plasma; GSH-Px, glutathione peroxidase; Hb, hemoglobin; SOD, superoxide dismutase.

**Table 5 antioxidants-14-01085-t005:** Multiple linear-regression analyses of the association of age and gender with γ-H2AX change (T_POST_ − T_BASE_).

Variables	Overall		A_NONE_		B_EXPOSURE_	
	Std. β	*p*	Std. β	*p*	Std. β	*p*
Age, years	0.02	0.88	−0.37	0.12	−0.03	0.89
Gender, female	−0.34	0.06	−0.39	0.10	−0.12	0.67

For each demographic variable, the association was assessed by multiple linear-regression analyses using γ-H2AX change (T_POST_ − T_BASE_) as the dependent variable.

## Data Availability

The datasets generated and/or analyzed during the study are available through the corresponding author upon reasonable request. Public and scientific inquiries should be directed to C.G.S.

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
