# Peer review of "Oxidative Stress-Mediated DNA Damage Induced by Ionizing Radiation in Modern Computed Tomography: Evidence for Antioxidant-Based Radioprotective Strategies"

_antioxidants, 2025, doi:10.3390/antiox14091085_

Round 1

Reviewer 1 Report

The study researches an important clinical concern using a novel in vivo approach. However, adjustments for baseline, clarification of statistical data, and expanded discussion on limitations and mechanisms are essential to enhance the impact of the paper.

Please address the following questions: 

1. Provide a more detailed discussion or statistical adjustment for the baseline differences in age and sex between the ANONE and BEXPOSURE groups, as these factors may influence radiosensitivity and oxidative stress responses.

2. Figure 1 lacks detail. Please specify the number per group and the corresponding CT time point. And in the main part of the paper, give the supplementary results.

3. Although γ-H2AX is a robust marker, please consider discussing the potential inclusion of additional markers, such as  ATM/ATR kinase activation, p53 phosphorylation, and apoptosis indicators, in the discussion to strengthen mechanistic insights.

4. Summarize your key findings (absolute and relative DNA damage increases) in your results. The concept of antioxidant-based radioprotection is promising. Please expand the discussion with the specific antioxidants (e.g., vitamin C, N-acetylcysteine) and ongoing clinical or preclinical trials.

Reviewer 2 Report

1. Demographic Differences Between Groups:

While the study has an excellent structure and measurement methodologies, making it "robust," the specific composition of the groups in terms of age and sex introduces a potential confounding variable that, although the study attempts to minimise concerning the ICRP, may require further statistical exploration to confirm that the observed effects are, in fact, primarily attributable to CT exposure and the resulting oxidative stress, and not to underlying demographic differences. Therefore, the following must be answered:

Question 1: Were subgroup analyses or statistical adjustments performed to explore or mitigate the potential confounding effect of age and sex on the results of oxidative stress and DNA damage biomarkers? Was it considered whether these demographic variables could influence baseline antioxidant levels, which were already different between groups?

Furthermore, it is suggested that the information in Table 1 should not be repeated; it repeats exactly the same thing in the text.

2. Mechanism of "Mitigation" in TPOST:

It is mentioned that "A relative increase in antioxidant enzyme activity after exposure (in the BEXPOSURE group) mitigated previously observed differences in these biomarkers (Table S3)." It then states that "Unlike TBASE, glutathione peroxidase (P < 0.15) and superoxide dismutase (P < 0.28) levels in the patient group did not differ from those in the healthy volunteer group." This suggests normalisation, but "mitigated" could imply an active protective effect or a reduction of the initial difference.

Question 2. Could you explain the nature of this "mitigation" in more detail? Does this indicate that the antioxidant response was so robust that it offset the baseline differences between groups in some enzymes, or simply that the initial difference became less pronounced but the damage response remained active?

3. Markers of DNA Damage and Oxidative Stress:

The study relies primarily on γ-H2AX as a marker of DNA damage and on the OI biomarker panel. It is mentioned that "other markers [e.g., activation of ataxia-telangiectasia-mutated (ATM) and ATM and Rad3 (ATR)-related kinases, ATM/ATR-mediated p53 phosphorylation, and apoptosis] could provide more detailed information."

Question 3: Given the importance of understanding the complete mechanism, were samples collected for future exploration of these other markers, or was the study limited to only those mentioned? Are any follow-up studies planned to analyse these biomarkers in more detail?

4. Immediate Clinical Implications:

While the study justifies future interventions, it does not explore whether, in light of the findings of significant DNA damage even at low doses, any immediate or precautionary clinical recommendations can be derived (beyond the already known dose reduction).

Question 4: Considering the increased DNA damage, should radiologists or treating physicians take additional precautions in using CT in specific populations (perhaps with lower baseline antioxidant capacity, if it could be identified) or in the frequency of scans?

5. Variability in Antioxidant Response:

The interquartile ranges and standard deviations suggest considerable variability in individual responses. For example, the relative change in DNA damage in BEXPOSURE is mentioned as ranging from [-15, 60] to [40 ± 72].

Question 5: Were the reasons for this variability in the response to oxidative stress and DNA damage among exposed patients explored? Were there correlations with BMI, age, sex, or other baseline parameters?

Already reported in the previous question

Reviewer 3 Report

The manuscript by Ramos et al. entitled "Oxidative Stress-Mediated DNA Damage Induced by Ionizing Radiation in Modern Computed Tomography: Evidence for Antioxidant-Based Radioprotective Strategies” describes analysis of DNA damage, oxidative stress and antioxidant enzymes activity in blood samples from patients undergoing two phase abdominal-pelvis computed tomography. Although the article addresses the important issue of potential hazard associated with radiation exposure during CT scans, major improvements are need before it can be accepted for publication in the "Antioxidants”.

1) The quality of the English language should be improved throughout the manuscript.

2) Page 2, lines 56-57: “IR-induced DNA damage mediated by oxidative stress (OS) was described, which also accounts for the majority (60 to 70%) of the total IR-induced DNA damage” – this is true for low-LET radiation but not necessarily for high-LET radiation.

3) Page 2, lines 61-62: “If these species are not repaired or are repaired incorrectly, they cause structural chromosomal abnormalities…” – The sentence is unclear and misleading, as it incorrectly uses the term “repaired” in reference to reactive oxygen species (ROS). ROS cannot be repaired! DNA damage caused by ROS can be repaired.

4) Methods such as determination of F2-isoprostane level, FRAP, activity of glutathione peroxidase, superoxide dismutase, and catalase should be described in more details. On the other hand, the level of detail provided regarding data storage is excessive and may be reduced for conciseness.

5) Page 4, lines 145-148: “Box and whisker plots were constructed to illustrate differences in overall survival (OS) and DNA damage biomarkers between the interventional groups. Spearman´s and linear regression analyses were performed to assess multivariate correlations and associations.” – There are no box and whisker plots in the manuscript! There are no Spearman’s correlations and linear regression reported in the results section!

6) There are significant differences between Anone and Bexposure groups in age, gender, benign abdominal lesions and abdominal infection/inflammation. It is highly possible that these differences have significant impact on the results. This is confirmed by data in Table 2 where significant differences between groups are observed in GSH-Px, SOD, CAT. This shows that study design is not optimal.

7) Page 5, lines 180-181: should be “Table A3” instead of “Table S3”.

8) There is not much sense in comparison of exposed patients to unexposed control donors. Especially when significant differences between these two groups are present even before irradiation. Description of results should be focused on differences induced by radiation i.e comparison of marker levels in Bexposure group in Tbase and Tpost. These results should be shown as a box and whiskers plots as it is easier for the reader than the Tables. Table 3 is difficult to understand. Presenting the same data with focus on differences between Tbase and Tpost in Bexposure would greatly improve clarity of data presentation.

9) Page 7, line 221: “A CT-associated IR exposure showed a correlation between antioxidant activation and DNA damage in the BEXPOSURE group” – The authors should clearly indicate the table or figure that demonstrates this correlation. In its current form, the data presentation does not clearly show the correlation.

10) Table A2: The effective dose for each patient should be reported, rather than only the mean value for the entire group. Additionally, it would be of interest to examine whether there is a correlation between effective dose and biomarker levels.

Reviewer 4 Report

There are some comments.

There are statistically significant differences in age and sex between the control (ANONE) and exposure (BEXPOSURE) groups. These differences are likely to influence the results.
Therefore, it is necessary to have a balanced selection of control and exposure groups.

Blood samples were collected at TBASE-min 0 and TPOST-min 60 to measure biomarkers of oxidative stress (OS) and DNA damage.
It would be helpful to explain why the measurement was taken 60 minutes later.
Ideally, biomarkers would be measured several times at intervals.

A more detailed account of the ionizing radiation dose results should be provided.

It would be better to edit the Tables to improve readability, with emphasis on essential values.

Round 2

Reviewer 3 Report

I am satisfied with answers to my remarks and changes made to the manuscript. In my opinion the manuscript can be accepted for publication.

I have no other comments.

Reviewer 4 Report

The manuscript has been well revised.

Regarding the tables, it would be better to check the  consistency in the statistical reporting methods and formatting across all tables.